# Enhanced Sensitivity of Microring Resonator-Based Sensors Using the Finite Difference Time Domain Method to Detect Glucose Levels for Diabetes Monitoring

**Lilik Hasanah [1], Harbi Setyo Nugroho [1], Chandra Wulandari [1], Budi Mulyanti [2], Dilla Duryha Berhanuddin [3],*, Mohamad Hazwan Haron [3], P. Susthitha Menon [3], Ahmad Rifqi Md Zain [3,4],*, Ida Hamidah [5], Khairurrijal Khairurrijal [6] and Rizalman Mamat [7]**

[1]   Department of Physics Education, Universitas Pendidikan Indonesia (UPI), Bandung 40154, Indonesia; lilikhasanah@upi.edu (L.H.); harbisetyo@student.upi.edu (H.S.N.); chandrawulandari@student.upi.edu (C.W.)

[2]   Department of Electrical Engineering Education, Universitas Pendidikan Indonesia (UPI), Bandung 40154, Indonesia; bmulyanti@upi.edu

[3]   Institute of Microengineering and Nanoelectronics (IMEN), Universiti Kebangsaan Malaysia (UKM), Bangi 43600, Malaysia; hazwanharon9@gmail.com (M.H.H.); susi@ukm.edu.my (P.S.M.)

[4]   4213 John McKay Lab, John A. Paulson Building, 31 Oxford St, Harvard University, Cambridge, MA 02138, USA

[5]   Department of Mechanical Engineering Education, Universitas Pendidikan Indonesia (UPI), Bandung 40154, Indonesia; idahamidah@upi.edu

[6]   Physics of Electronic Materials Research Division, Faculty of Mathematics and Natural Sciences, Institut Teknologi Bandung, Bandung 40132, Indonesia; krijal@fi.itb.ac.id

[7]   Department of Mechanical Engineering, Universiti Malaysia Pahang, Pekan 26600, Malaysia; rizalman@ump.edu.my

*   Correspondence: dduryha@ukm.edu.my (D.D.B.); rifqi@ukm.edu.my (A.R.M.Z.); Tel.: +60-012-442-2292 (D.D.B.); +60-019-573-8877 (A.R.M.Z.)

**Abstract:** The properties of light and its interaction with biological analytes have made it possible to design sophisticated and reliable optical-based biomedical sensors. In this paper, we report the simulation, design, and fabrication of microring resonator (MRR)-based sensors for the detection of diabetic glucose levels. Electron Beam Lithography (EBL) with 1:1 hydrogen silsesquioxane (HSQ) negative tone resist were used to fabricate MRR on a Silicon-on-Insulator (SOI) platform. Scanning Electron Microscopy (SEM) was then used to characterize the morphology of the MRR device. The full-width at half-maximum (FWHM) and quality factors of MRR were obtained by using a tunable laser source (TLS) and optical spectrum analyzer (OSA). In this paper, the three-dimensional Finite Difference Time Domain (3D FDTD) approach has been used to simulate the proposed design. The simulation results show an accurate approximation with the experimental results. Next, the sensitivity of MRR-based sensors to detect glucose levels is obtained. The sensitivity value for glucose level detection in the range 0% to 18% is 69.44 nm/RIU. This proved that our MRR design has a great potential as a sensor to detect diabetic glucose levels.

**Keywords:** microring resonator (MRR); quality factor; sensitivity; three-dimensional Finite Difference Time Domain (3D FDTD); diabetes detection

## 1. Introduction

Microring resonators (MRRs) are vital photonic components in the development of high-index silicon photonic platforms. It produces optical resonance by utilizing the constructive interference phenomenon from the interaction between propagated light in its waveguides [1–3]. The MRR waveguide consists of a ring and a straight waveguide that interact in the presence of light propagation, provided that the distance or gap between them is small, usually in micro or nanometer sizes [4,5]. They are highly sought after due to several advantages such as fast modulation, enabling spectral filtering, compact sizes, and flexibility in structures. Silicon is the material of choice to fabricate MRRs, as advanced silicon technology, especially in electronic industries, has made it possible to design MRRs at a relatively low cost [6]. Recent developments in the field of MRRs have led to a renewed interest in optical sensing applications, especially in the biomedical field [7]. It has been reported that MRRs have the ability to produce an integrated sensing device with high sensitivity that allows detection of ultra-low molecule concentration [8,9]. Moreover, there are reports on label-free MRRs based sensors, which makes them highly attractive as advanced biomedical instrumentations, either invasive or noninvasive [10–12]. MRR-based sensors also have applications in various fields such as clinical analysis, substance detection in food, health, and environment monitoring [13–15].

Diabetes is one of the major health problems globally, with more than 400 million people affected [16]. Diabetes is a condition of excessive glucose levels in the blood [17]. Glucose, $C_6H_{12}O_6$, is classified as a carbohydrate from the monosaccharide subcategory, which is the most important energy source for metabolism processes [18,19]. Diabetes is often closely associated with a variety of other medical conditions such as celiac disease, cystic fibrosis, tuberculosis, and heart attacks [20]. Complications from various types of diseases caused by diabetes can lead to blindness, kidney failure, amputation, cardiovascular diseases, and even cancer [21]. A study conducted by the American Diabetes Association (2017) states that diabetes can be diagnosed from the amount of glucose in the blood, as shown in Table 1 [22]. A person can be diagnosed with diabetes if there is more than 0.13% of glucose in the blood. In this work, the focus is to measure the sensitivity of MRR-based sensors in detecting the glucose range of diabetics. Thus, the detected glucose levels are in the range of 0.13–0.40%, with changes in glucose levels of 0.09%. Although the glucose percentage in a diabetic person is relatively low, it is crucial to have a system that has the ability to detect its presence in order to prevent the adverse effects that may come later.

**Table 1.** Category of diabetes diagnosis.

| Category | Concentration of Glucose in the Blood When Fasting (%) |
| --- | --- |
| Healthy | 0.07–0.11 |
| Prediabetes | 0.11–0.13 |
| Diabetes | 0.13 |

Previous research on MRR-based sensors has only focused on the performance of sensors for glucose detection [23–25]. In this paper, we report the design and characterization of MRR-based sensors to detect and subsequently predict diabetic glucose levels. The mechanism of MRR-based sensors occurs on its surface, which then provided specific processing. This specific processing occurs due to the analyte binding on the surface element, which leads to refractive index changes and a resonance shift. The presence of glucose molecules at very low levels can still change the value of the refractive index so that it causes a shift in resonance. Thus, MRR-based sensors have a high potential to be used as a low-concentration glucose detection sensor. Initially, a sensitivity simulation of MRR-based sensors was performed to detect the glucose levels in diabetic patients. The MRR device was then fabricated using Electron Beam Lithography (EBL) and a negative tone hydrogen silsesquioxane (HSQ) resist mask. Characterization of the device was carried out by using an optical spectrum analyzer (OSA) and tunable laser source (TLS) to measure the quality factor, Q. After that, a quality factor simulation was carried out using the 3D Finite Difference Time Domain (FDTD) method,

which was then verified with the experimental data. These results were then utilized to predict glucose levels by observing the MRR resonance shift.

## 2. Materials and Methods

### 2.1. Design Consideration and 3D FDTD Approach

Figure 1a shows the proposed design of the MRR waveguide. It consists of a ring waveguide coupled to two straight waveguides, thus enabling them to interact with each other within a small distance (gap). Based on MRR's geometric configuration (Figure 1b), the proposed design consists of a ring radius (R), gap (g), waveguide height (h), and waveguide width (W), with values as shown in Table 2.

**Table 2.** The dimensions of the proposed microring resonator (MRR).

| MRR Geometry | Dimension (µm) |
| --- | --- |
| Ring radius | 4.50 |
| Gap | 0.05 |
| Waveguide width | 0.50 |
| Waveguide height | 0.22 |

A ring radius of 4.50 µm is considered to have the ability to reduce bending loss, while a waveguide cross-section of $0.50 \times 0.22$ µm$^2$ provides low propagation loss. In addition, a gap of 0.05 µm was chosen to minimize the impact of optical coupling scattering. The silicon waveguides were formed in 1 µm silica buffer layer and supported by a silicon substrate of 250 µm. The Silicon-on-Insulator (SOI) configuration of MRR (Figure 1b) was favored due to its ability to confine and, thus, optimize light propagation in waveguides. Based on the proposed design, the FDTD approach has been carried out. The Finite Difference Time-Domain method, or FDTD, is often used to simulate a photonics device because of its ability to provide both the spatial and temporal properties of photonics structures at once in a single calculation. FDTD uses Maxwell's equations based on Yee Mesh to derive numerical scattering problems and electromagnetic absorption on the basis of Maxwell's equations [26]. In this work, Lumerical FDTD Solutions software has been used to simulate the proposed design of SOI-based MMRs. Throughout this study, the three-dimensional (3D) FDTD method was used to simulate MRR based on SOI because it was able to provide good accuracy, though it was time and power consuming.

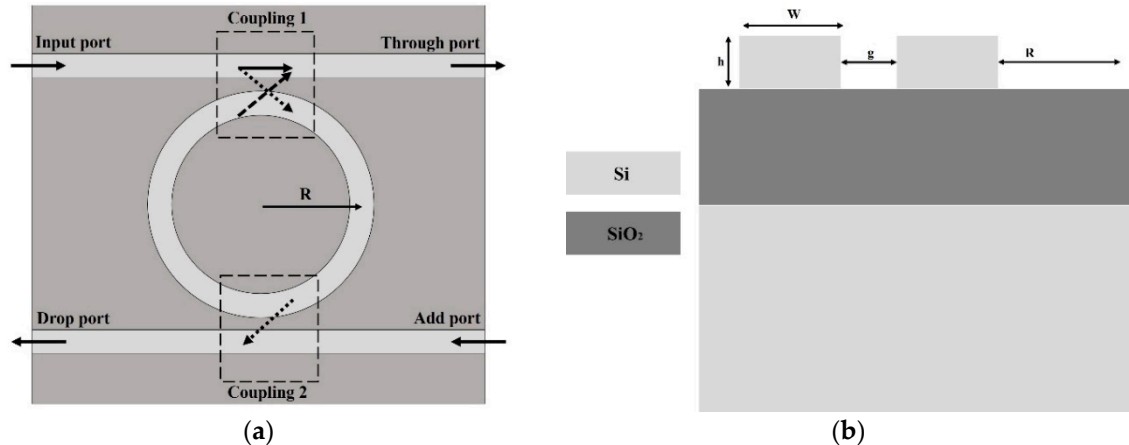

(a)　　　　　　　　　　　　　　　　　　　　　　　　　(b)

**Figure 1.** Illustration of the proposed design of the microring (MRR) shows (**a**) configuration of add-drop system ports and (**b**) the cross-section of the MRR based on Silicon-on-Insulator (SOI) material.

### 2.2. Fabrication Process and Device Characterization

The illustration of the fabrication process of MRR on SOI substrates is shown in Figure 2. MRR silicon waveguides were fabricated from the Silicon-on-Insulator (SOI) wafer layer using Elionix F125 Electron Beam Lithography (EBL) for pattern writing and the Hydrogen Silsesquioxane (HSQ) negative tone resist mask in the etching process for pattern transfer. The HSQ resist mask was diluted with the ratio of 1:1 in methyl isobutyl ketone (MIBK). The HSQ resist mask then spun at 3000 rpm for 60 s. The sample was then baked at 90 °C for 40 min. Next, the electron beam from the EBL system was exposed throughout the sample to write the MRR waveguides. The exposed patterns were then developed using tetramethylammonium hydroxide (TMAH) with a concentration of 25% in water for 30 s. The etching process was done using an Surface Technology System - Reactive Ion Etch (STS-RIE) machine with a combination of $SF_6$ and $C_4F_8$ for 3 min. The residual of HSQ resist mask was cleaned from the sample surface using isopropyl alcohol (IPA). The fabricated MRR device consisted of a single ring waveguide and two straight waveguides, called the add-drop microring resonator system.

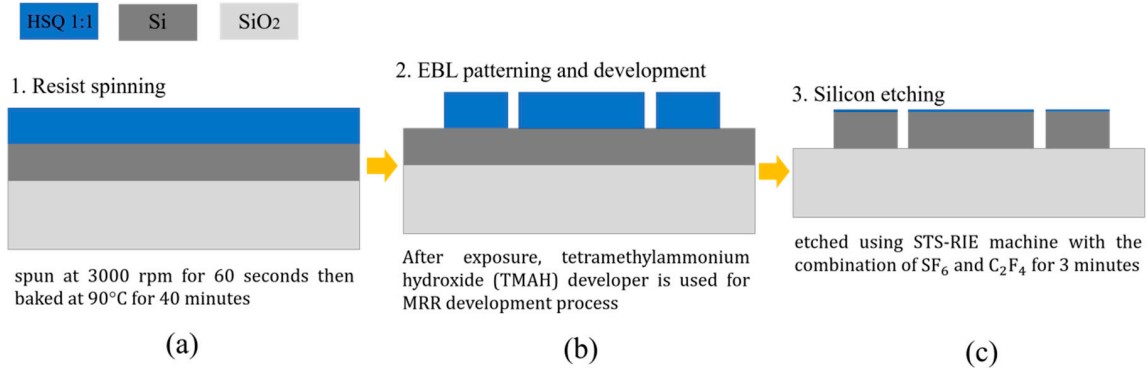

**Figure 2.** Three steps of the device fabrication process as follows: (**a**) resist spinning, (**b**) Electron Beam Lithography (EBL) patterning and development, and (**c**) silicon etching.

Figure 3 shows the experimental setup for MRR based on SOI quality factor characterization. The surface morphology of the MRR device was characterized using Scanning Electron Microscopy (SEM) to observe several morphological parameters such as ring radius, gap, and waveguide width. The fabrication process successfully fabricated the MRR device with good repeatability. Good dimensional control of the MRR device was proven with morphological parameters. Subsequently, the value of the quality factor was characterized using an ANDO AQ4321D Tunable Laser Source (TLS) and Agilent 86142B Optical Spectrum Analyzer (OSA). TLS supplies the light to the MRR device, which covers wavelengths from 1525 to 1575 nm. The optical spectrum analyzer (OSA) was used to measure output power at the through-port of MRR device. The value of the quality factor is calculated by using Equation (1) [27];

$$Q = \frac{\lambda_{res}}{FWHM} \tag{1}$$

where $\lambda_{res}$ is the resonance wavelength, and FWHM is value of half of maximum intensity between two extreme points of the resonance curve.

### 2.3. Diabetes Detection by FDTD Approach

The investigation of MRR-based sensors for diabetes detection has been carried out by 3D FDTD using Lumerical FDTD Solutions. The FDTD method in Lumerical FDTD Solutions was used because of its ability to perform good and accurate analyses of light propagating in MRR waveguides. The dimensions of the MRR in this simulation were based on the proposed design of MRRs based on SOI, as shown in Table 2.

In order to obtain the desired results that are near the experimental results, we have optimized several parameters in this simulation. The simulation time was set to 50,000 fs, which is 10 times higher than the initial simulation time, to obtain a more detailed calculation from the FDTD method. The mesh type was set to the automesh feature with the configuration of the mesh being nonuniform to minimize the effect of numerical dispersion. The mesh accuracy was set to 3 in order to get a good trade-off between accuracy, memory requirements, and the total time needed. As for the perfectly matched layer boundary condition, the standard profile was chosen as it will produce a good overall light absorption with a relatively small number of layers so that the required simulation time can be reduced. As a start, the simulation was run for the index background of 1 (air) in the environment temperature of 295 K. This condition was chosen to imitate the real conditions of the experimental work for quality factor measurement. Figure 4 shows the computer-aided design of the MRR device from Lumerical FDTD Solutions.

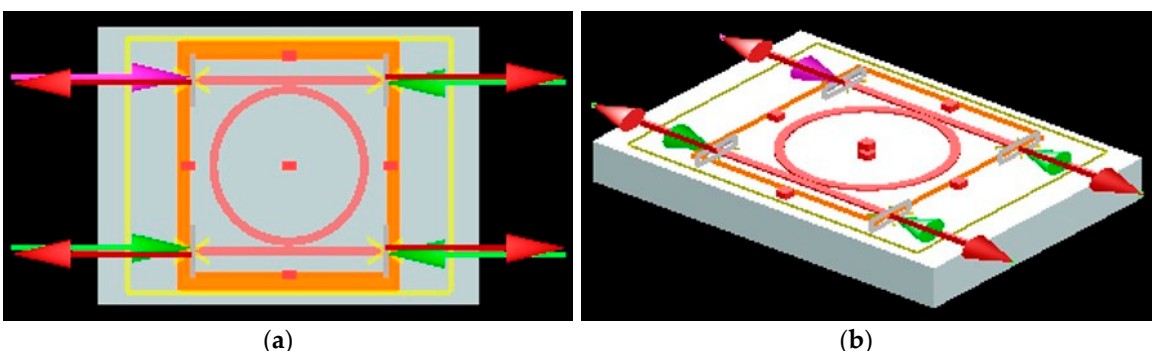

**Figure 3.** Experimental setup for quality factor characterization in MRR waveguides by through-port power transmission measurements using an optical spectrum analyzer (OSA) and tunable light source (TLS) as a light source.

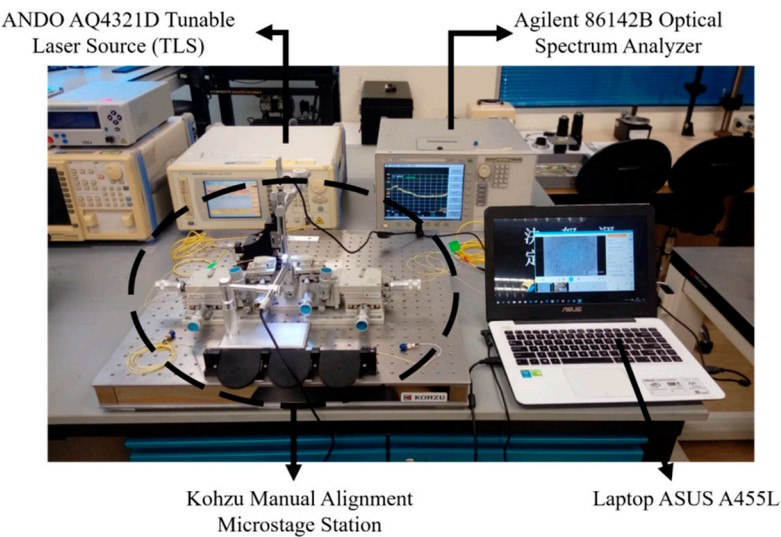

| (a) | (b) |

**Figure 4.** The geometry of the designed MRR in this simulation work from (**a**) top view and (**b**) bird's-eye view.

Figure 5 shows a schematic on how the occurrence of the glucose analyte will cause a resonance shift to the MRR-based sensor. The addition of glucose in the environment around the MRR waveguide triggers an increase in the environmental refractive index due to the appearance of glucose analytes. The presence of glucose analytes interferes with the evanescent field, as they will absorb energy initially possessed by the field [28]. If the evanescent field experiences loss of energy due to being absorbed by the glucose analyte, then the total energy of the light traveling on the waveguide will

decrease. This is caused by the evanescent field, which is part of the light propagation mode on the MRR waveguide. The energy loss will also decrease the speed of the light phase to propagate on the waveguide. By slowing down the phase and speed of light, the effective waveguide index is affected, as the effective refractive index of the waveguide itself can be defined as the ratio between the speed of the light phase in the vacuum and the speed of the light phase in the waveguide [29]. Based on these definitions, it can be concluded that the decrease in phase velocity will result in an increase in the effective refractive index of the waveguide, thus enabling the resonance shift.

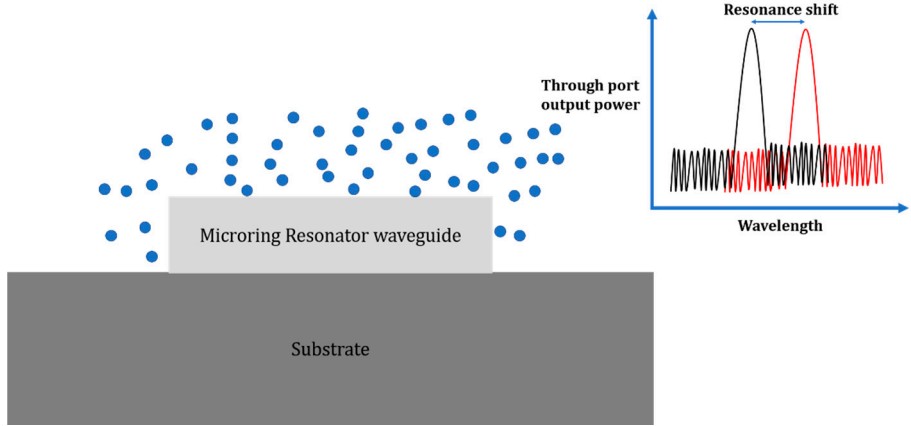

**Figure 5.** The schematic on how the resonance shift occurs due to the occurrence of glucose analytes on the MRR-based sensor.

Next, to predict the ability of the MRR-based sensor to detect diabetes, the resonance shift that occurs due to changes in glucose concentration in the environment around the MRR were observed. The glucose concentration in the MRR environment will be varied based on glucose levels in people with diabetes. It can be done using Lumerical FDTD Solutions by changing the index background of the simulation to the reference refractive index of certain glucose concentrations. Based on experimental data obtained through a study done by Yeh et al. [30], the refractive index of glucose at certain glucose concentrations can be obtained using Equation (2) as below;

$$n_{g/L} = a(\lambda).C + b(\lambda) \tag{2}$$

where C is the glucose concentration in units of g/L, $b(\lambda)$ is the refractive index of water or 0 g/L glucose concentration at a wavelength of 1550 nm, which has a value of 1.3101 [6], and $a(\lambda)$ is a wavelength-dependent constant with a value of $1.189 \times 10^{-4}$ [30]. Sensors based on MRR performance in detecting glucose or other biochemical molecules can be identified through their sensitivity values. MRR-based sensitivity values are defined mathematically in Equation (3) [31]. S is the sensor sensitivity, $\Delta\lambda_{res}$ is a resonance shift, and $\Delta n_L$ is a change in the environmental refractive index.

$$S = \frac{\Delta\lambda_{res}}{\Delta n_L} \tag{3}$$

## 3. Results and Discussion

### 3.1. Simulation Approach for Predicting the Experimental Result

Figure 6 shows the 3D FDTD simulation results of MRR transmission spectra at the through-port for the proposed design arrangement. It shows that the proposed design in this study had a transmission value of nearly 80% at a resonance peak wavelength of 1543.70 nm with FWHM and quality factors of approximately 0.456 nm and 3385.307 nm, respectively. In this study, we optimized the FDTD simulation using minimum mesh steps up to 0.25 nm and a simulation time of 50,000 fs. Additionally,

the standard Perfectly Matched Layer (PML) boundary conditions were used to provide good overall absorption. This optimization was done in order to achieve a balance outcome between accuracy, memory requirement, and simulation time.

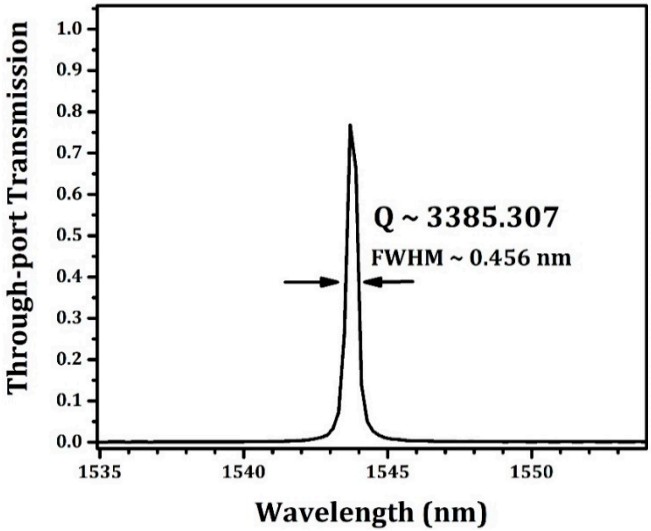

**Figure 6.** The simulation result of the transmission at the through-port of MRR for index background of air (n = 1).

### 3.2. Experimental Results

The surface morphology of the fabricated MRR was successfully captured using SEM, as shown in Figure 7. The fabrication process using the Elionix F125 EBL system for pattern writing and HSQ as a negative tone resist mask for pattern transfer successfully produced the morphology of MRR with good repeatability and accurate dimensional control. We observed that the fabricated MRR had a ring radius, gap, and waveguide width of $-4.35$, $-4.95 \times 10^{-2}$, and $-5.02 \times 10^{-1}$ μm, respectively. The differences in fabricated MRR compared to the proposed design were also considerably low: 3.13%, 0.10%, and 0.32% for ring radius, gap, and waveguide width, respectively.

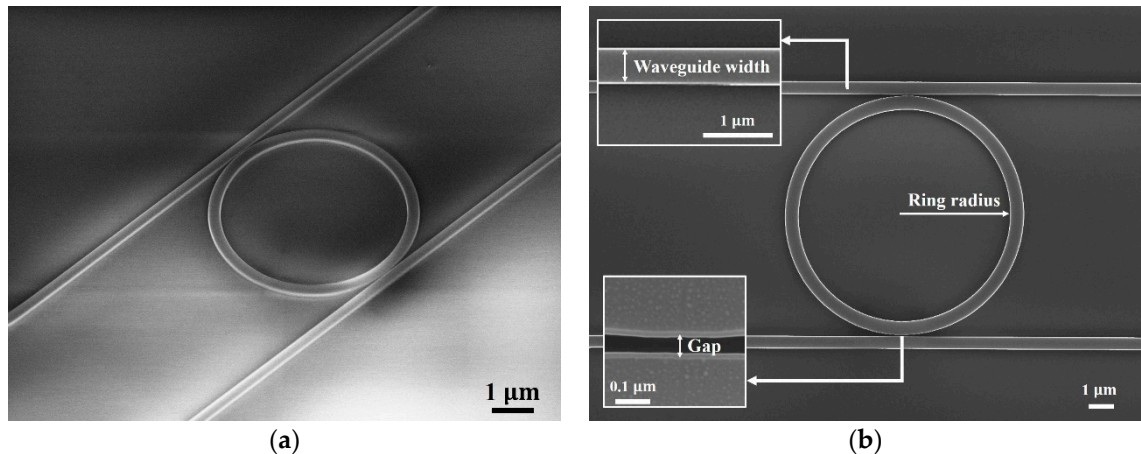

**Figure 7.** SEM image of fabricated MRR from (**a**) bird's-eye view and (**b**) top view.

Characterization of the quality factor value has been done by measuring the through-port power of MRR transmission using OSA, while TLS, which covers wavelengths in the range of 1525 to 1575 nm, was used to supply the light that propagates in the MRR waveguide. Figure 8 shows the measurement results of the MRR device's output power at the through-port point, including the value of the quality

factor and FWHM from each resonance peak of the different samples. Each device, namely samples 1–6, was fabricated using the same method and configuration. The purpose of measuring all samples is to show the repeatability and consistency of the fabrication process. The similarity between the value of the quality factor and FWHM from each resonance peak of the different samples has, once again, proven the consistency of dimensional control of the fabrication process.

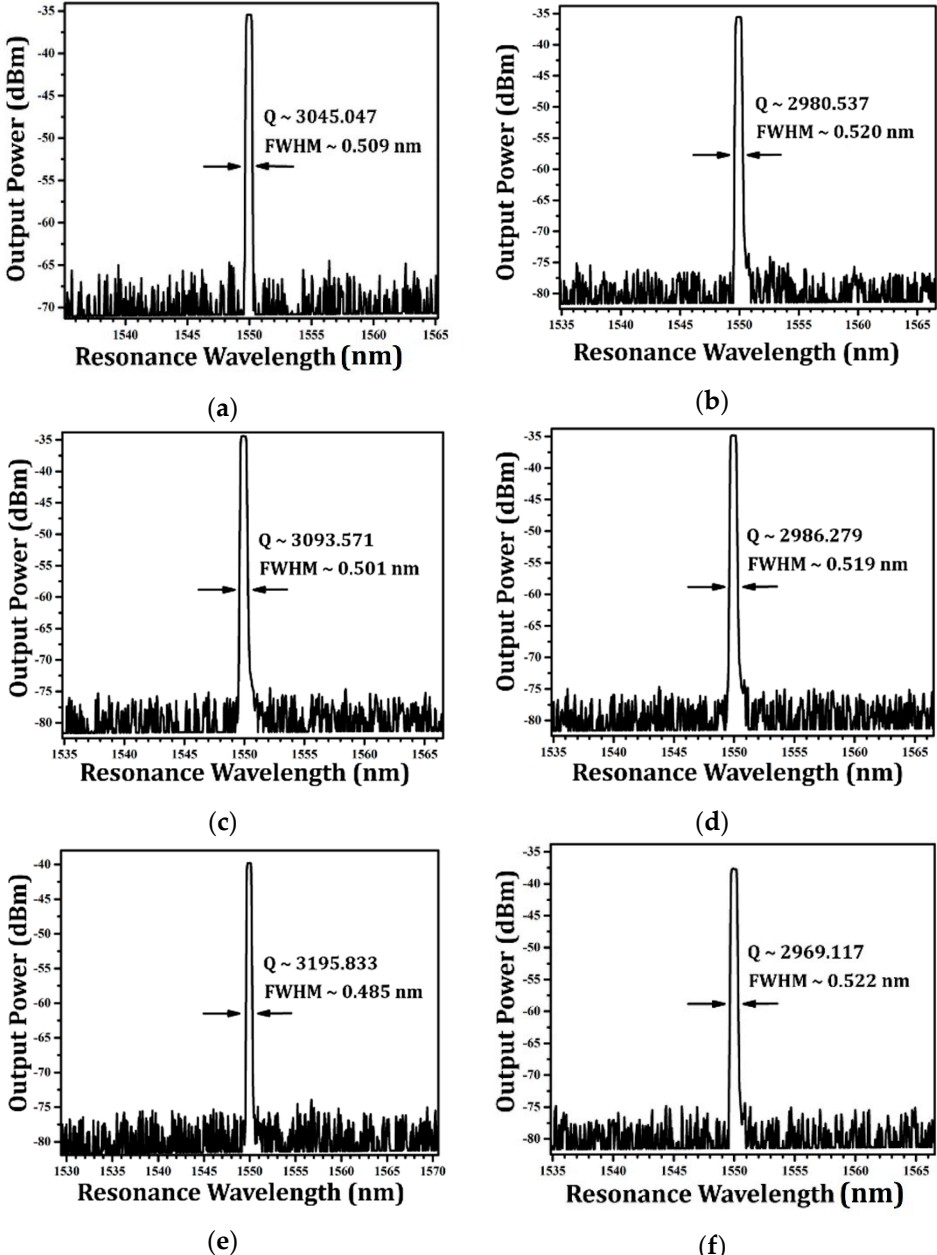

**Figure 8.** The measurement results at the through-port of the MRR device from (**a**) sample 1, (**b**) sample 2, (**c**) sample 3, (**d**) sample 4, (**e**) sample 5, and (**f**) sample 6. The values of quality factor, Q, and FWHM for each resonance peak are also shown.

The quality factor of the fabricated devices varied from 2969.117 nm to 3195.583 nm, with Sample 5 having the best values. There was a good agreement of the quality factor value between the experimental result (3195.583 nm), as shown in Figure 8e, and the simulation result (3385.307 nm) from Figure 6. Therefore, the 3D FDTD can be utilized to predict the sensitivity for MRR-based sensors, as it depends on the value of the quality factor [32].

### 3.3. Simulation Results for Diabetes Detection

The presence of glucose at different concentrations in the environment results in a resonance shift, as shown in Figure 9. This is due to the change in the effective refractive index of the waveguides with different concentrations of the analyte. The change in the effective refractive index will then trigger a resonance shift towards a longer wavelength, as shown in Equation (4) [33].

$$\lambda_{res} = 2\pi Rn_{eff}/m \tag{4}$$

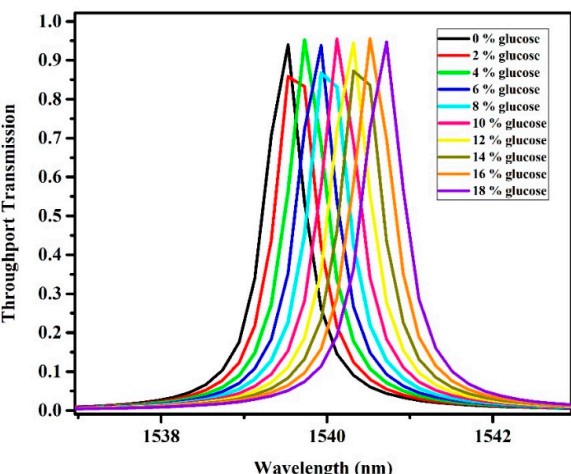

**Figure 9.** The resonance pulses obtained using the 3D Finite Difference Time Domain (FDTD) method approach showing resonance shifts for each glucose concentration.

Figure 10 shows a linear relation between the resonance shifts from MRR resonance pulses and changes in the environmental refractive index. This linearity was similar to previous studies of MRR-based sensors by Guider et al. and Ciminelli et al. [8,25]. The linear curve has a positive, linear slope, which means that the relationship between changes in the environmental refractive index are triggered by an increase in glucose concentration and the resonance shifts. This demonstrates that the addition of glucose around the MMR system will shift the resonance towards a larger wavelength, as theoretically described in Equation (4). The sensitivity of the MRR-based sensor obtained in this study was 69.44 nm/RIU.

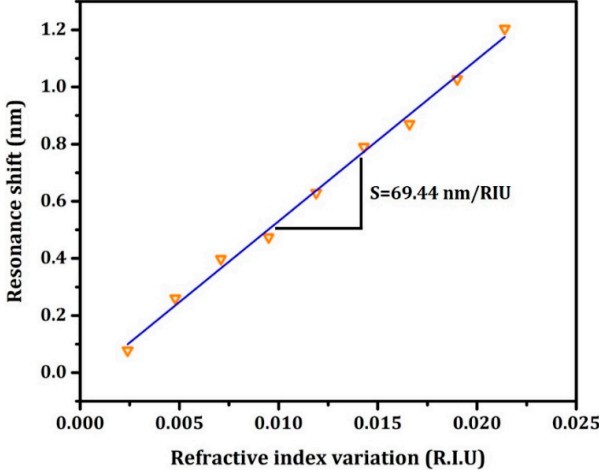

**Figure 10.** The linear curve of the relationship between resonance shifts is relative to changes in the environmental refractive index and sensitivity of the sensor based on MRR.

Figure 11 shows a response curve simulation of a sensor based on MRR on changes in glucose concentration. The changes are indicated by the shift in resonance that occurs. The tested glucose concentrations were based on the glucose level in the blood of diabetics, as shown in Table 1. A resonance shift was observed, which showed that MRR-based sensors were able to respond to changes in glucose concentration, which is very low in people with diabetes. This shows that MRR-based sensors have a great potential to be developed as an effective diabetes detection device. Despite small variations in the glucose concentration and resonance shift, the changes can still be detected optimally due to the sensitivity value of the sensor.

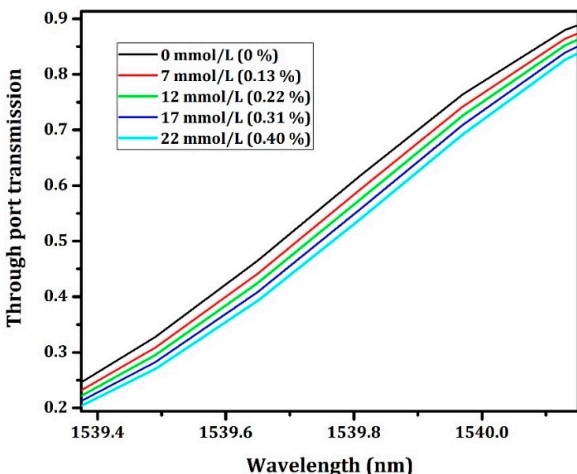

**Figure 11.** Resonance shift that occurs in MRR-based sensors due to the different glucose concentrations of a diabetic person.

## 4. Conclusions

We have successfully investigated the ability of an MRR-based sensor configuration to detect diabetes. MRR on the SOI platform has been fabricated using the EBL system and the HSQ negative tone as a resist layer. The morphology and quality factor of the fabricated MRR have also been experimentally characterized. The simulation value of quality factors is close to the experimental value. Therefore, the simulation method can be used and is sufficient to obtain the sensor's sensitivity. The resonance shifts were also observed with changes in diabetic glucose concentration. Sensor sensitivity values can still be optimized in order to detect changes at even lower glucose concentrations. Optimization of the MRR-based sensor sensitivity can be done by adjusting the size of the MRR geometric parameters so that it has a higher quality factor value, as there is linearity between the value of the quality factor and the sensitivity of the MRR-based sensor. Optimization by adjusting the size of the MRR geometric parameters can be done using a 3D FDTD simulation first by varying the size of the MRR geometric parameters to predict the value of the quality factor. The overall results will advance development in diabetes monitoring technology, especially in non-invasive sensing devices. Coupled with real-time detection and label-free features, MRR-based sensors will contribute to the increase in effectiveness while reducing operating costs in diabetes monitoring technology.

**Author Contributions:** Conceptualization, B.M., L.H., and R.M.; methodology, I.H. and K.K.; software, B.M.; H.S.N. and C.W.; validation, L.H., D.D.B., P.S.M., and A.R.M.Z.; formal analysis, H.S.N.; investigation, H.S.N., C.W., and M.H.H.; resources, A.R.M.Z.; data curation, L.H., D.D.B., and H.S.N.; writing—original draft preparation, H.S.N.; writing—review and editing, L.H., D.D.B., and P.S.M.; visualization, H.S.N.; supervision, L.H., B.M., D.D.B., and P.S.M.; project administration, B.M.; funding acquisition, B.M. All authors have read and agreed to the published version of the manuscript.

**Funding:** We acknowledge the Directorate of Research and Community Service, Ministry of Research, Technology and Higher Education, Republic of Indonesia, for financial supporting through the grant of Penelitian Terapan Unggulan Perguruan Tinggi. The authors also acknowledge the Ministry of Education of Malaysia for financial support under the Fundamental Research Grant Scheme FRGS/1/2015/TK04/UKM/02/4.

**Acknowledgments:** We acknowledge the administrative support given by Institute of Microengineering and Nanoelectronics (IMEN), UKM, and Universitas Pendidikan Indonesia (UPI) while doing this research.

**Conflicts of Interest:** The authors declare no conflicts of interest. The funders had no role in the design of the study; in the collection, analyses, or interpretation of data; in the writing of the manuscript, or in the decision to publish the results.

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
