# Peer review of "Enhanced Sensitivity of Microring Resonator-Based Sensors Using the Finite Difference Time Domain Method to Detect Glucose Levels for Diabetes Monitoring"

_applsci, doi:10.3390/app10124191_

Round 1

Reviewer 1 Report

This manuscript described a microring resonator-based sensor to detect Glucose Level for Diabetes Monitoring. The author details the design consideration and production process. The sensitivity value of MRR-based sensor for glucose levels detection is ranged from 0 to 18 %. The author suggested that MRR design has a great potential as a sensor to detect diabetic glucose level.

Here, I have minor considerations are cited below.

Diabetes can be diagnosed from the amount of glucose in the blood, according to Table 1, the concentration of glucose in the blood of healthy is 0.07 – 0.11%, Pre-diabetes is 0.11 – 0.13% and Diabetes is 0.13%. Fig. 11 shows the different glucose concentration level of diabetic person from 7mmol/L (0.13%) to 22mmol/L. How about the glucose concentration is lower than 7mmol/L? Currently, there are quite a few instruments and methods for detecting blood glucose, and the detection range of general blood glucose meters can already reach 0.6 to 33.3 mmol / L. What is the advantage of MRR-based sensor for glucose detection? P.8, Fig.8, what is the difference between sample1~sample6?

Reviewer 2 Report

The work “Enhanced Sensitivity of Microring Resonator-Based Sensor Using Finite Difference Time Domain Method to Detect Glucose Level for Diabetes Monitoring et. al. is of potential interest for journal readers. This work can be accepted for publication after revision.

English needs revision. Grammar and language of the manuscript should be revised. Small sentences should be preferred. Authors can compare the results with other resonator based sensors for glucose to show the superiority of this work. What real sample used to for glucose detection?  Conclusion should include the important inferences of the work. It is better to remove the numerical values in the conclusion as they can be given in abstract but not in conclusion. Moreover, conclusion should also state the importance/future implications of the work.

Round 2

Reviewer 2 Report

Authors have improved the manuscript but there are still few shortcomings which can be addressed before publication.

Comments:

If authors corrected the English language of the whole manuscript, why it is not highlighted on the manuscript? Please revise the English language of the manuscript and please highlight the corrections. Conclusion should also state the importance/future implications of the work which means what is the effect of your work on the field of glucose sensing? And how it will change the future technology in this field? Please include in Conclusions section.
